# An Additive Manufacturing Direct Slicing Algorithm Based on a STEP Model

Xingguo Han [1,*], Zhuangchao Zhan [2], Xiaohui Song [1] and Lixiu Cui [1]

[1] School of Mechanical Engineering, Guilin University of Aerospace Technology, Guilin 541004, China; sxhui@guat.edu.cn (X.S.); cuilixiu@guat.edu.cn (L.C.)

[2] School of Mechanical and Electrical Engineering, Guilin University of Electronic Technology, Guilin 541004, China; zhan_zhuang_chao@163.com

[*] Correspondence: hanxingguo2004@163.com

**Abstract:** The Standard Template Library (STL) file is the most common data format for the description of an additive manufacturing (AM) geometric model, but it has some disadvantages, such as large errors of the geometric model description, the easy loss of topology information, data duplication, large file sizes, and so on. Aiming at these problems, a direct slicing algorithm based on a Standard for the Exchange of Product Model Data (STEP) model was proposed. For the parts composed of basic types of surfaces such as boundary curves, spherical surfaces and cylindrical surfaces, the traditional geometric method was used to calculate the intersection. For the parts with complex surfaces, the three-dimensional models were described based on Non-Uniform Rational B-Spline (NURBS) surfaces. The NURBS surfaces were layered using a discrete tracking algorithm, the tracking starting point was determined, the intersection line between the tangent plane and each NURBS sub-surface was obtained, and the closed layer contour was formed. Finally, the slicing simulations and printing experiments of solid parts were carried out using the direct slicing algorithm based on the STEP model. It was shown that the dimensional accuracy and surface quality of the printed parts from this algorithm had been significantly improved.

**Keywords:** additive manufacturing; direct slicing; complex surface; NURBS; geometric method

## 1. Introduction

Additive manufacturing (AM), also known as 3D printing, is a set of manufacturing technologies that is capable of building parts layer upon layer [1]. AM integrates computer-aided design (CAD) technology, motion control technology and material forming technology, so as to realize the processing and manufacturing of the parts [2]. In recent years, AM technology has played an essential role in the fabrication of customized products at a low cost. Some countries have successively nurtured the AM industry as one of the most important industries of the future [3].

### 1.1. Fused Deposition Modeling (FDM)

FDM is one of the most popular and ubiquitous AM methods because of its rapid production, cost-efficiency, ease of access, broad material adaptation, and capability to produce complex components [4,5]. FDM uses filament as its feedstock material. Such filaments can be manufactured by specific extruding machines or purchased from commercial brands [6]. The filament is pushed into the liquefier through a mechanism, heated to a molten state, extruded with a nozzle, then adhered to the formed layer. When the printing of one layer has been completed, the z-axis moves a certain distance equal to layer thickness in order to print the next layer, until the formation of the whole part is completed [7,8]. The principle of AM technology is shown in Figure 1.

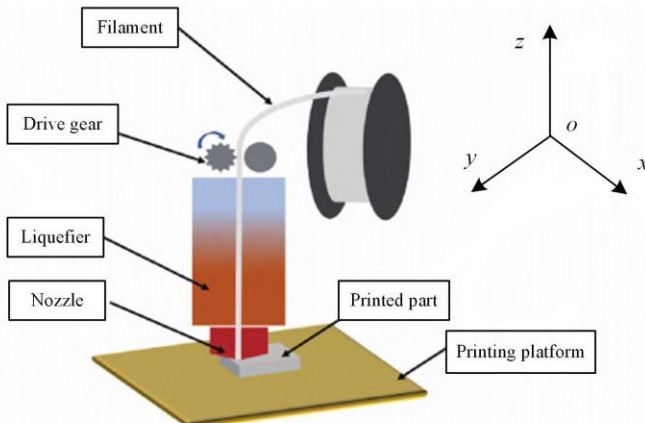

**Figure 1.** Schematic diagram of FDM.

The materials used in FDM are mainly polymers such as nylon, acrylonitrile butadiene styrene (ABS) and polylactic acid (PLA). Other possible materials for FDM are polyetheretherketone (PEEK), polypropylene (PP), polyphenylsulphone (PPSF), and composite filaments. Many materials have been developed and studied, and their properties—such as their mechanical, thermal and electrical properties—have been reported [9–11].

Conventional printers generally use a Cartesian structure or a parallel structure, as shown in Figure 2. The printing temperature of such printers is below 300 °C, which requires a low forming temperature of materials. When the filament is deposited through the nozzle layer by layer to form a part, the bonding strength between two layers affects the mechanical properties of the part. Because the materials used in FDM also need to have a certain melt strength, the adhesion between layers should be good, so as to avoid cracking caused by thermal stress or a low melt viscosity between layers [12–14].

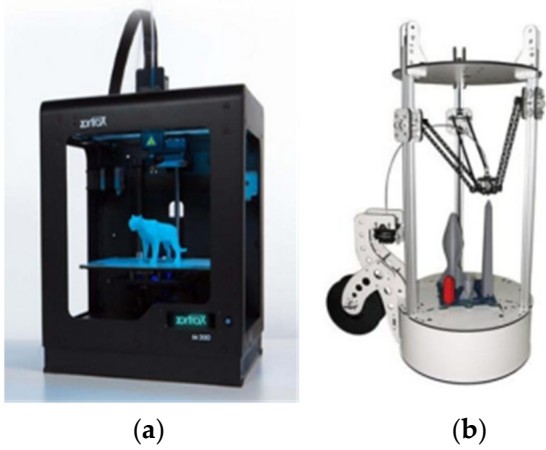

(**a**)                    (**b**)

**Figure 2.** AM devices. (**a**) Cartesian structure; (**b**) parallel structure.

*1.2. Slicing Algorithm*

The method of discretizing the AM model according to layer thickness along the slicing direction is called the slicing algorithm, which is the soul and foundation of AM [15]. In the slicing processing, the tangent plane parallel to the XY plane is generally used to cut the three-dimensional (3D) solid model, and the layer contour position information of each layer is obtained [16]. The 3D manufacturing is changed into manufacturing layer by layer in the two-dimensional (2D) plane, which simplifies the manufacturing process. The accuracy and quality of the layer contour position information obtained by slicing ultimately affect the accuracy and quality of AM products [17,18]. AM slicing technology mainly includes a slicing algorithm based on a Standard Template Library (STL) model and a direct slicing algorithm based on a CAD model [19,20].

### 1.2.1. Slicing Algorithm Based on a Standard Template Library (STL)

As we all know, the STL format is the mainstream in AM for the depiction of a 3D model, which uses many triangular meshes as the boundaries of the model. As the most widely used numerical control programming language, machine control code (GCode) [15] is also the interactive language between almost all AM printers and computers or controllers. Additionally, GCode is the main programming language of slicing algorithms based on an STL model, and can be directly generated by CAD software or open-source slicer software, and so on [20–22].

For the slicing algorithm for the STL format, some scholars established the topological relationship of triangular patches in order to make slicing easier and more efficient. Each layer plane could intersect with its adjacent triangular patch to generate one or two intersections generally. In each layer plane, all of the intersections were connected end to end to form a closed curve or polygon. Besides this, some researchers presented some slicing strategies based on an STL model for the rapid construction of topology information. Such slicing strategies also improved the efficiency of mesh simplification, and established the chart of adjacency of triangular patches in order to improve the searching efficiency and slicing speed [23–25].

Slicing algorithms based on the position information of triangular patches were studied. The triangular patches were classified and sorted according to the Z value of the vertex. In the slicing process, the intersection relationship of some triangular patches was determined by comparing the height of the layer plane with the Z value of the patch, so as to generate the contour line. Some scholars used the special function in MATLAB software to calculate the intersections between the tangent plane and the triangular patch in order to generate the contour layer by layer, and carried out the example simulation. However, the algorithm has some difficulties in practical application [26–28].

In terms of adaptive slicing, an adaptive slicing algorithm was presented based on an STL model, which could automatically identify the internal and external closed curves of each layer, and improve the slicing efficiency [29–31]. Some researchers studied a slicing algorithm with adjustable layer thickness according to the surface curvature of the parts to be processed, which effectively improved the forming quality of the parts. Besides this, another slicing algorithm presented could automatically adjust the layer thickness according to the real-time change of the contour of each layer, so as to obtain better surface quality and dimensional accuracy [32,33].

Some academics introduced a real-time slicing algorithm to improve the accuracy of the layer contour, analyzed the error of the layer contour, and proposed a solution to improve the accuracy of the slicing contour. For the layer contour distortion of the parts with complex curved surfaces, the curve fitting method based on double arc interpolation was used to obtain the accurate section contour and analyze its position error [34,35].

In short, the slicing algorithms based on STL use a series of triangular patches to approximate the surface contour of a CAD model. These algorithms have advantages like simple calculation and high efficiency, and are supported by most commercial AM systems, such that these kinds of algorithms have become a standard in some common commercial software or open-source slicers. However, these algorithms also have some disadvantages, such as large errors of geometric model description, the easy loss or error of topology information, data redundancy, large file sizes, greater slicing times of large parts, and so on.

### 1.2.2. Direct Slicing Algorithm Based on a CAD Model

The direct slicing algorithm can avoid the above problems of layering with STL files. As for the direct slicing, some scholars have adopted the point cloud data of a CAD model in order to layer parts. Lee et al. developed a method to generate a part layer contour from three-dimensional model point cloud data, and developed an approach to simplify the number of point clouds [36–38]. Chen et al. proposed to generate point cloud data by scanning 3D solid parts. The point cloud data intersected with layered planes in order to directly generate layer contour curves [39]. Zhao and Cao developed a direct slicing

algorithm based on AutoCAD and C$^{++}$ language, which used the intersection function in AutoCAD to obtain the intersection lines between a 3D entity and tangent planes, and output them according to the corresponding data format [40,41].

According to the convex–concave characteristics of different areas of additive manufacturing parts, Wang et al. used a different layered thickness to directly slice the CAD model, which reduced the printing time by about 30–40% without affecting the surface quality [42]. Sun et al. developed direct slicing software for a CAD model on the solid model kernel Parasolid of Unigraphics in C language [43]. Hope et al. proposed a direct slicing algorithm using IGES files, and compiled a piece of slicing software [44,45].

Some scholars have worked on a direct slicing algorithm for the Standard for the Exchange of Product Model Data (STEP) model. Zhou et al. proposed a direct slicing algorithm based on a STEP model, studied the extraction method of the geometric topology information of the STEP model, adopted the geometric intersection algorithm between the tangent plane and the basic curve or surface, and obtained the intersection line of the parametric surface and tangent plane [46].

In summary, the above direct slicing algorithms can improve the slicing accuracy of parts to a certain extent, reduce the surface error of parts, and avoid errors of STL files. Meanwhile, these algorithms can also reduce the pre-processing time of AM and save the error diagnosis and repair procedures. However, there are some disadvantages, such as their complex calculations and huge files. Meanwhile, for most of the above algorithms, each section contour is still a linear approximation of the solid section. Especially for some complex surface parts, the slicing accuracy and efficiency need to be further improved. Therefore, the direct slicing algorithm has become a very popular research direction in AM technology.

### 1.3. Our Work and Its Application Significance

### 1.3.1. Our Work

This paper presents a direct slicing algorithm for the STEP model based on a Non-Uniform Rational B-Spline (NURBS) surface. Given the characteristics of the geometric elements and surface types of the STEP model, different intersection algorithms were selected in slicing processing. For the parts with complex surfaces, a direct slicing algorithm based on a NURBS surface and STEP model was proposed. The NURBS surface was layered by a discrete tracking algorithm to construct a closed layer contour. For the parts with basic types of surfaces—such as boundary curves, spherical surfaces, cylindrical surfaces and so on—the traditional geometric method can be used to calculate the intersection, so as to finish slicing. This method can save program calculation time and improve the efficiency and robustness of the algorithm.

### 1.3.2. Industrial Utilization and Application Significance

Slicing technology is important as one of the AM operation steps. The main purpose of this algorithm is to obtain the boundary line with higher accuracy in each layer. Meanwhile, it has higher computational efficiency, and can quickly, directly and accurately convert the design ideas into prototypes, or manufacture parts. Therefore, this algorithm has high industrial utilization and application significance, which are mainly reflected in the following aspects:

(1)　AM technology is highly valued by academia and industry, and is widely used in the manufacturing of aerospace parts, mold manufacturing, medical devices and organ manufacturing, personalized customization, building construction, and many other fields. In these fields, many parts have complex surfaces, such as the blades of aeromotors, human bones, and so on. However, some traditional slicing algorithms or general commercial software cannot ensure the printing accuracy of these parts. In this paper, the STEP geometric model is transformed into NURBS surface representation, and a closed layer contour with higher accuracy is constructed. This slicing algorithm reduces the surface shape error of parts and improves printing accuracy.

(2)    AM technology can obviously shorten the research and development cycle of the product. Generally speaking, the higher the slicing accuracy of parts, the smaller the layer thickness and the greater the amount of calculation. This leads to data redundancy, large storage files and longer slicing times. The slicing algorithm in this paper can effectively shorten the product development cycle, reduce the product development cost, and improve the production efficiency.

(3)    Usually, the functions of most commercial slicing software are fixed. Therefore, AM printer users are only able to select slicing patterns according to their requirements, and cannot carry out secondary development. The freedom of choosing slicing methods is severely restricted because the commercial software is only based on planar slicing, and does not include other functions such as curved layer slicing. However, the slicing algorithm in this paper can solve these problems, and has high portability and flexibility.

(4)    AM is a green manufacturing mode. Some solid parts are designed as hollow network structures, and it is difficult to produce them by traditional processing methods. However, AM can not only print complex parts with a hollow network structure or other complicated structures but also save materials and reduce the pressure on the natural environment and resources.

## 2. Method

### 2.1. Expression Method of a Basic Boundary Line

As for the STEP model, the basic boundary lines construct a three-dimensional model, mainly including a straight line and a quadratic curve (circle, ellipse, hyperbola, etc.) [47]. These boundary lines can be described in implicit form and parametric form.

The implicit expression of the straight line is as follows:

$$ax + by + c = 0 \tag{1}$$

where $x$ is the abscissa variable, $y$ is the ordinate variable, and $a$, $b$ and $c$ are constants.

The parametric expression of the straight line is as follows:

$$P(t) = C + t\vec{d} = 0 (t \in R) \tag{2}$$

where $C$ represents the local coordinate system origin, $\vec{d}$ represents the direction of this line, and $t$ is a parameter.

The quadratic curve can be expressed in implicit form, as follows:

$$a_{00}x_0^2 + 2a_{01}x_0x_1 + a_{11}x_1^2 + b_0x_0 + b_1x_1 + c = 0 \tag{3}$$

where $A = [a_{ij}]$ is a $2 \times 2$ matrix, $B = [b_i]$ and $X = [x_i]$ are $2 \times 1$ vectors, then Equation (3) can be denoted as follows:

$$X^\mathrm{T}AX + B^\mathrm{T}X + c = 0 \tag{4}$$

The parametric equation of the quadratic curve can be expressed as:

$$\begin{cases} x = x(t) \\ y = y(t) \end{cases} \quad (t \in [a,b]) \tag{5}$$

A circle, ellipse, hyperbola and parabola can be expressed by the above quadratic curve equation, in which a point and straight line are the special cases of a quadratic curve. In slicing processing, a three-dimensional expression is often used to describe the boundary line, which is treated as a three-dimensional graphic element [48]. The expressions of a straight line, ellipse, hyperbola and parabola in three-dimensional space are deduced as follows:

$$P(t) = C + (r_1 \cos t)\,\hat{u} + (r_2 \sin t)\hat{v} \tag{6}$$

$$P(t) = C + (r_1 \sec t)\,\hat{u} + (r_2 \tan t)\hat{v} \tag{7}$$

$$P(t) = C + F\left(t^2\hat{u} + 2t\hat{v}\right)\vec{d} \tag{8}$$

where Equations (2) and (6–8) represent expressions of a straight line, ellipse, hyperbola and parabola, respectively; $u$ and $v$ represent the two different coordinate axes; $r_1$ and $r_2$ represent the constant value on the respective coordinate axis; and $F$ is the focus of the parabola.

### 2.2. Intersection Algorithm of the Basic Boundary Line and Tangent Plane
2.2.1. Intersection of the Line and Tangent Plane

According to the definition of the STEP model in AP203, which is an application protocol of the STEP format, the straight line belongs to a directed line segment [44]. In order to calculate the intersection between a tangent plane and a straight line, we can define the tangent plane as follows:

$$ax + by + cz + d = 0 \tag{9}$$

where $a^2 + b^2 + c^2 = 1$, and $|d|$ represents the minimum distance from the coordinate origin to the plane $\pi$. The vector $\hat{n} = [a\ b\ c]$ is normal for the plane $\pi$, as shown in Figure 3.

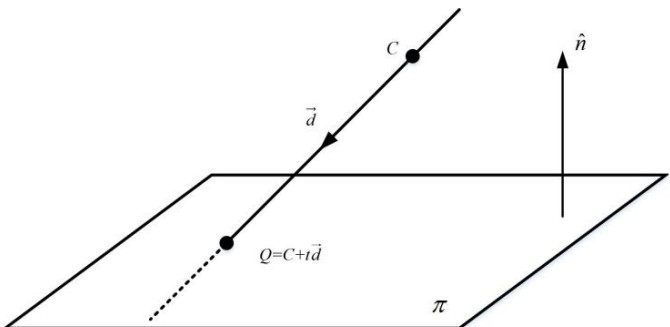

**Figure 3.** Intersection between a straight line and a plane.

Assuming that a straight line and a plane intersect at a point $Q$, the intersection $Q$ satisfies Equations (2) and (9). Equation (2) is substituted into Equation (9) as follows:

$$a(C_x + d_x t) + b(C_y + d_y t) + c(C_z + d_z t) + d = 0 \tag{10}$$

According to Equation (10), the parameter $t$ is given as

$$t = \frac{aC_x + bC_y + cC_z + d}{ad_x + bd_y + cd_z} \tag{11}$$

Equation (11) is converted into the vector form as

$$t = \frac{-(\hat{n}\cdot C + d)}{\hat{n}\cdot\vec{d}} \tag{12}$$

In Equation (12), $\hat{n}\cdot\vec{d}$ denotes the dot product between the direction vector of the straight line $l$ and the normal vector of the tangent plane $\pi$. If $\hat{n}\cdot\vec{d} = 0$, it means that the straight line $l$ is parallel to the plane $\pi$. If the straight line parallel to plane $\pi$ is in this plane, there are countless intersections. If the straight line is not in plane $\pi$, there is no intersection. Parameter $t$ can be obtained according to Equation (12), and the intersection coordinates can be found by substituting parameter $t$ into Equation (2).

2.2.2. Intersection of the Tangent Plane and Quadratic Curve

As shown in Figure 4, the vector method is adopted in the intersection operation of a tangent plane and quadratic curve. When the tangent plane $\pi$ and a spatial curve $P$ intersect at point $M$, $O$ is the coordinate origin of the spatial coordinate system and $\hat{n}$ is the unit normal vector of plane $\pi$. A vertical line of plane $\pi$ is made through the origin $O$ and intersects at point $L$. According to the principle of vector addition [49], the relationship among the vectors ($\overrightarrow{OM}$, $\overrightarrow{OL}$ and $\overrightarrow{LM}$) is given as follows:

$$\overrightarrow{OM} = \overrightarrow{OL} + \overrightarrow{LM} \tag{13}$$

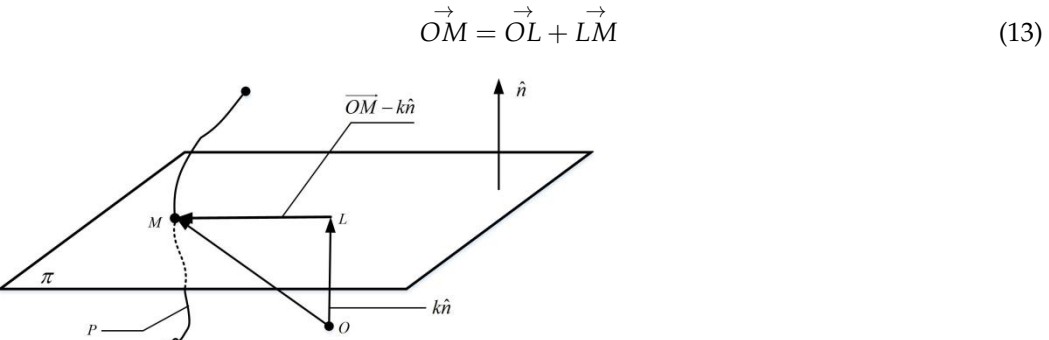

**Figure 4.** Intersection schematic diagram of the tangent plane and boundary line based on the vector method.

Assume that the distance from coordinate origin $O$ to the tangent plane $\pi$ is $k$, then $\overrightarrow{OL} = k\hat{n}$. Because vector $\overrightarrow{OL}$ is perpendicular to tangent plane $\pi$, the dot product of $\overrightarrow{LM}$ and $k\hat{n}$ is zero, as follows:

$$\left( \overrightarrow{OM} - k\hat{n} \right) \cdot k\hat{n} = 0 \tag{14}$$

Because intersection point $M$ belongs to the boundary curve $P$ of which the curve equation is determined, the coordinate value of the intersection point $M$ can be obtained by combining Equation (14) with the equation of the boundary curve $P$.

*2.3. Research on a Direct Slicing Algorithm of the STEP Model Based on a NURBS Surface*

The basic type surface of a STEP model mainly includes a plane and a quadric surface, and belongs to the bounded region with direction. The traditional geometric intersection algorithm can be used to find the intersection between the tangent plane and basic type surface. For complex surface parts, the direct slicing algorithm based on NURBS surface can be adopted.

2.3.1. Overview of the NURBS Surface

A NURBS surface is a general standard tool for 3D model representation, and was selected as the only method to exchange general splines, single-segment polynomials, and rational curve and surface splines by STEP. In STEP files, solid and surface models can be represented by NURBS surfaces [50], as follows:

$$S(u,v) = \frac{\sum\limits_{i=1}^{n+1}\sum\limits_{j=1}^{m+1} W_{ij} P_{ij} b_{ik}(u) b_{jl}(v)}{\sum\limits_{i=1}^{n+1}\sum\limits_{j=1}^{m+1} W_{ij} b_{ik}(u) b_{jl}(v)} \tag{15}$$

where $u$ and $v$ represent the parameters in the $u$ direction and $v$ direction, respectively; their numerical values are between 0 and 1. $m$ and $n$ represent the number of parameters in the $u$ direction and $v$ direction, respectively. $P_{ij}$ is the control vertex of the NURUS surface, and $W_{ij}$ is the weight factor. $b_{ik}(u)$ represents the $k$-degree B-spline basis function in the $u$ direction, and $i$ is the number of functions. $b_{jl}(v)$ represents the $l$th order B-spline basis

function in the $v$ direction, and $j$ is the number of functions. The B-spline basis functions can be obtained by Cox-deBoor recurrence formula [49], as follows:

$$b_{j1}(s) = \begin{cases} 1 & ku_j \leq u \leq ku_{j+1} \\ 0 & \text{others} \end{cases} \tag{16}$$

$$b_{l1}(s) = \begin{cases} 1 & kv_l \leq v \leq kv_{l+1} \\ 0 & \text{others} \end{cases} \tag{17}$$

$$b_{jk}(u) = \frac{(u - ku_j)b_{(j)(k-1)}(u)}{ku_{j+k-1} - ku_j} + \frac{\left(ku_{j+k+1} - u\right)b_{(j+1)(k-1)}(u)}{ku_{j+k+1} - ku_{j+1}} \tag{18}$$

$$b_{lk}(v) = \frac{(v - kv_l)b_{(l)(k-1)}(v)}{kv_{l+k-1} - kv_l} + \frac{(kv_{l+k+1} - v)b_{(l+1)(k-1)}(v)}{kv_{l+k+1} - kv_{l+1}} \tag{19}$$

In Equations (16–19), $ku_j$ and $kv_l$ represent the node vectors of the NURBS surface in the $u$ direction and $v$ direction, respectively.

The STEP file contains the data information required for NURBS surfaces. Reading this information from the STEP file requires the transformation of the EXPRESS format of geometric model feature data in STEP/AP203 into the data storage format of the $C^{++}$ class, establishing the standard interface of $C^{++}$ relative to the STEP file, and completing the reading of geometric feature information.

### 2.3.2. NURBS Surface Intersection Method Based on a Discrete Tracking Algorithm

The main problem in layering based on a NURBS surface is to find the section intersection between the tangent plane and the three-dimensional model; the discrete method [51] and the tracking method [52] are effective algorithms to solve this problem. The discrete method is simple and reliable, but it has low accuracy and needs to store a large amount of data. Although the tracking method has high calculation accuracy and lower data storage, it is not easy to select the tracking starting point, and the stability of this method is poor. Aiming at the intersection problem between the NURBS surface and tangent plane, this paper proposed a discrete tracking algorithm. Because there was a certain topological relationship among the sub-surface patches of the STEP model, this topological relationship was used to judge the next sub-surface patch to enter. The discrete tracking algorithm has the advantages of simple calculation, a lack of complex iterative operation, and high calculation accuracy. The steps are:

Step 1: Firstly, the NURBS surface is divided into several small sub-surfaces, which are numbered one by one.

Step 2: Judge whether each sub-surface meets the flatness requirements one by one. The user defines the flatness and sets its error range. The sub-surface is regarded as a plane if it meets the requirements. Otherwise, the sub-surface will continually be equally divided into four sub-surfaces until the flatness requirements are met. Each sub-surface, by quartering, meets the flatness requirements and a quadtree structure is formed.

Step 3: Determine the starting point of tracking. If the intersection line between the NURBS surface and tangent plane is unclosed, take the intersection of the NURBS surface boundary line and tangent plane as the starting point. If the intersection line is a closed intersection line ring, judge the intersection between the polyhedron bounding box and the tangent plane of the NURBS surface. In the event of intersection, it indicates that there is an intersection line ring. Divide the NURBS surface into many sub-surfaces, and then obtain the intersection points of their boundary curves and tangent plane, respectively. These intersection points are used as the tracking starting points of the intersection line ring.

Step 4: Calculate the intersection between the sub-surface and the tangent plane. Because the sub-surface meets the flatness requirements, we can calculate the intersection line between the sub-surface (or plane) and the tangent plane, and judge the next adjacent

sub-surface entering the intersection line according to the topological relationship of the sub-surfaces.

Step 5: Repeat Step 4 until the intersection line reaches the boundary of the NURBS surface or returns to the tracking starting point. Connect all of the intersection points in turn to form the intersection line between the NURBS surface and the tangent plane.

## 3. Algorithm Steps

For the direct slicing algorithm based on a STEP model, the geometric model feature data from the STEP/AP203 file should first be obtained, and should then be converted into the data storage format based on C$^{++}$ class in the slicing system. Secondly, we should determine the intersections of various types of surfaces and boundary lines in the model. The layer contour data required for additive manufacturing can be obtained last. The specific steps are as follows:

Step 1: Import the STEP model of the part and extract the geometric feature data described based on EXPRESS in the model.

Step 2: Transform the EXPRESS description of the geometric feature data in the STEP model into the data storage format based on C$^{++}$ class, and establish the standard data exchange interface between the C$^{++}$ language and STEP file in order to complete the reading of the geometric feature information.

Step 3: Determine the slicing thickness $\Delta z$ and the layering direction of additive manufacturing.

Step 4: Calculate the minimum and maximum coordinate values of the surface along the layering direction in the STEP model.

Step 5: Adopt different intersection algorithms given the diversity of printed part surfaces. For the basic types of surfaces such as boundary curves, spherical surfaces and cylindrical surfaces, the traditional geometric intersection algorithm is adopted. If the surface of the printed part is complex, the NURBS surface intersection algorithm based on a discrete tracking algorithm is used.

Step 6: Connect the intersection points or intersection lines in turn to generate the layer contour.

Step 7: Increase the height of the tangent plane by $\Delta z$, and then judge whether its height exceeds the maximum coordinate value along the layering direction of the model. If it doesn't exceed the limit, continue to perform Step 5 in order to carry out the intersection operation of the next layering plane. Otherwise, perform Step 8.

Step 8: Finish the layered processing and save the contour data information of each layer.

## 4. Experiments

### 4.1. Slicing Simulations and Printing Experiments

The STEP model of the lower teeth, as shown in Figure 5a, was sliced. Because this model contains many complex surfaces, the direct slicing algorithm based on a NURBS surface and STEP model was adopted. The STEP model of the lower teeth was described by the NURBS surface. This model was sliced by a discrete tracking algorithm to determine the tracking starting point, obtain the intersection line between the tangent plane and each NURBS sub-surface, and construct the layer contour of each layer. The layer thickness was set as 0.3 mm, and the layer's direction was along the positive direction of the Z axis. A slicing simulation of the lower teeth based on the STEP model was carried out, as shown in Figure 5b.

The STEP model of the gearbox base, as shown in Figure 5c, was sliced. Because the model was composed of basic surfaces, the traditional geometric method was used, which could make full use of the geometric characteristics of the basic surfaces. The layer thickness was 0.3 mm, and the layer direction was along the positive direction of the Z axis. The slicing simulation of the gearbox base based on the STEP model was carried out as shown in Figure 5d.

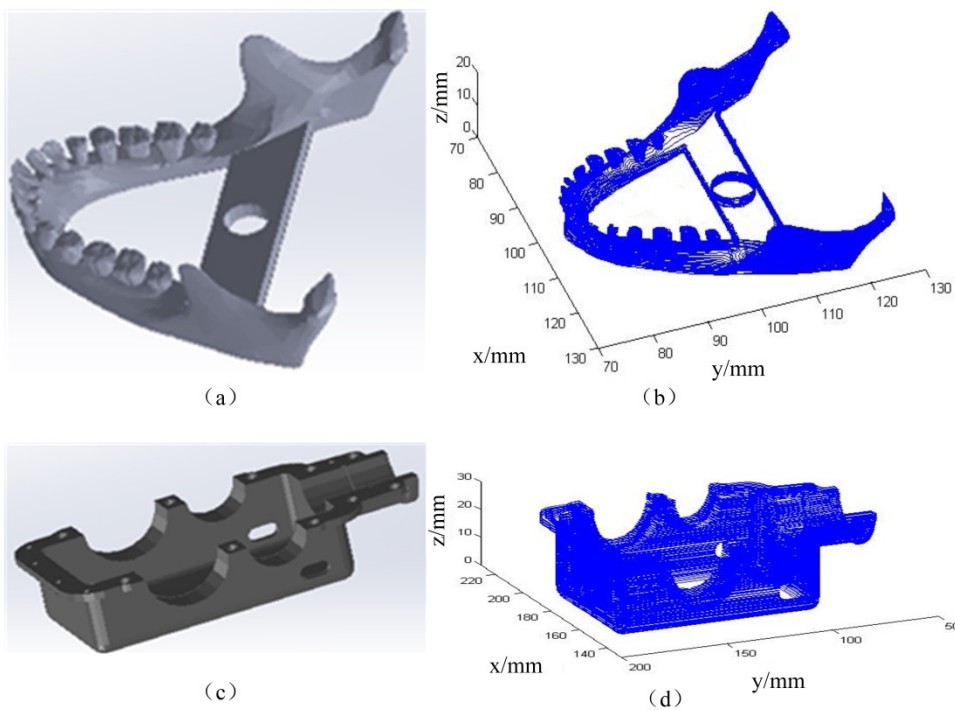

**Figure 5.** STEP modes and their slicing simulation results. (**a**) STEP model of the lower teeth; (**b**) slicing simulation of the lower teeth; (**c**) STEP model of the gearbox base; (**d**) slicing simulation of the gearbox base.

In order to verify the feasibility and effectiveness of the research on the direct slicing algorithm of AM based on the STEP model, the printing experiment was carried out on a cantilever AM platform, as shown in Figure 6. The parts shown in Figure 5a,c were printed, respectively. The solid printed parts are shown in Figures 7 and 8, and the print parameters are shown in Table 1.

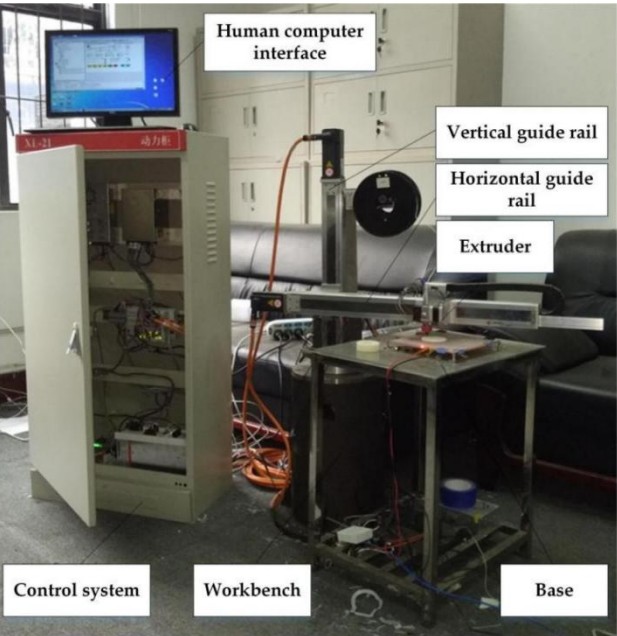

**Figure 6.** The cantilever AM platform.

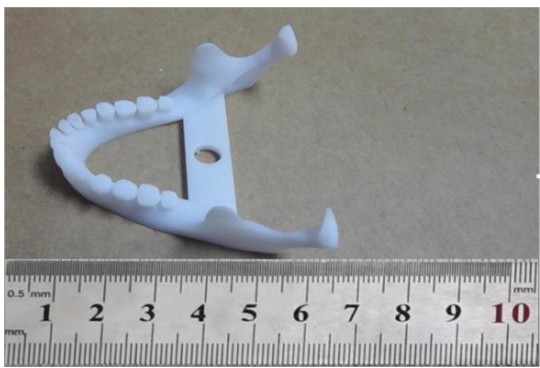

**Figure 7.** The printed lower teeth.

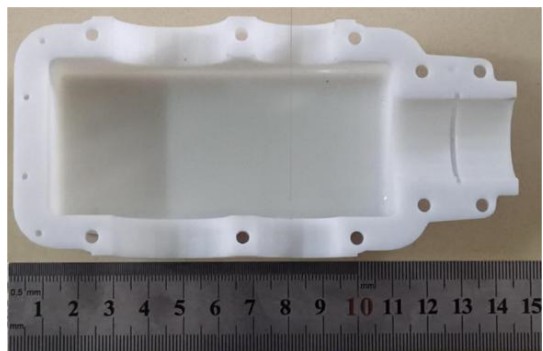

**Figure 8.** The printed gearbox base.

**Table 1.** Print parameters.

| Printed Part | Print Material | Layer Thickness (mm) | Total Layers | Nozzle Diameter (mm) | Nozzle Temperature (°C) | Filling Density | Extrusion Speed (mm/s) |
|---|---|---|---|---|---|---|---|
| Lower teeth | ABS | 0.1 | 196 | 0.1 | 215 | 100% | 60 |
| Gearbox base | ABS | 0.2 | 149 | 0.2 | 230 | 100% | 100 |

*4.2. Result and Discussion*

The surface roughness of the lower teeth was measured on the platform of a surface roughness meter, as shown in Figure 9. The arithmetic mean deviation of the profile ($R_a$) and the maximum height of the profile ($R_z$) were used as the evaluation parameters of the surface roughness. For FDM technology, the parts are formed by stacking layer by layer using the hot melting and adhesion of thermoplastic print materials. With the increase of the print height, the side contour and shape of the printed parts will change slightly under the influence of gravity. Therefore, the surface roughness of their sides is slightly higher than that of their upper surface. For the printed lower teeth, ten positions on the upper surface and side, respectively, were selected. The surface roughness of these positions was measured, as shown in Table 2. In order to better evaluate the surface roughness objectively, the sample's mean, standard deviation and coefficient of variation (also called the relative standard deviation) were used for evaluation.

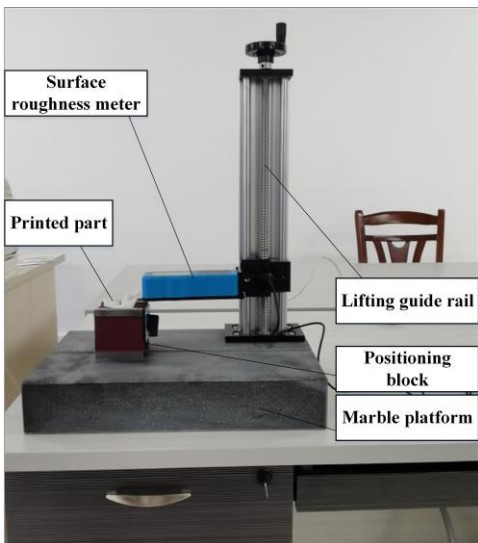

**Figure 9.** Platform of the surface roughness meter.

**Table 2.** Measured surface roughness of the lower teeth.

| Location Number (Upper Surface) | $R_a$ (μm) | $R_z$ (μm) | Location Number (Side) | $R_a$ (μm) | $R_z$ (μm) |
|---|---|---|---|---|---|
| 1 | 7.214 | 32.217 | 11 | 11.528 | 50.124 |
| 2 | 6.925 | 31.114 | 12 | 12.577 | 49.387 |
| 3 | 8.312 | 28.368 | 13 | 14.026 | 47.219 |
| 4 | 8.473 | 28.259 | 14 | 13.780 | 52.54 |
| 5 | 7.511 | 27.11 | 15 | 11.451 | 47.324 |
| 6 | 8.160 | 33.872 | 16 | 12.134 | 49.671 |
| 7 | 7.207 | 32.931 | 17 | 13.572 | 45.216 |
| 8 | 6.698 | 27.458 | 18 | 12.980 | 51.317 |
| 9 | 8.219 | 29.743 | 19 | 13.247 | 48.216 |
| 10 | 7.835 | 30.117 | 20 | 11.879 | 53.335 |

$$\overline{X} = \frac{1}{n} \sum_{i=1}^{n} X_i \tag{20}$$

$$S = \frac{1}{\sqrt{n-1}} \sqrt{\sum_{i=1}^{n} \left(X_i - \overline{X}\right)^2} \tag{21}$$

$$V = \frac{S}{\overline{X}} \times 100\% \tag{22}$$

where $\overline{X}$, $S$ and $V$ represent the sample mean, sample standard deviation and coefficient of variation respectively. $n$ is the total amount of samples, $i$ is the number of samples, and $X_i$ is the $i$th sample value.

According to Equations (20–22), the sample mean, sample standard deviation and coefficient of variation of $R_a$ measured on the upper surface of the printed lower teeth were 7.655 μm, 0.630 μm and 8.23%, respectively, and the sample mean, sample standard deviation and coefficient of variation of $R_z$ were 30.119 μm, 2.361 μm and 7.84%, respectively.

The sample mean, sample standard deviation and coefficient of variation of $R_a$ measured on the side of the printed lower teeth were 12.717 μm, 0.943 μm and 7.41% respec-

tively, and the sample mean, sample standard deviation and coefficient of variation of $R_z$ were 49.435 μm, 2.529 μm and 5.12%, respectively.

However, when the traditional slicing algorithm was used to print the lower teeth, the sample means of $R_a$ and $R_z$ measured on the upper surface were 12.322 μm and 45.275 μm, respectively. The sample means of $R_a$ and $R_z$ measured on the side of the printed lower teeth were 20.296 μm and 65.014 μm, respectively. It is obvious that the surface quality of the printed part using the algorithm in this paper was improved compared with the traditional slicing algorithm.

The length, width and height of the gearbox base were measured 10 times, as shown in Table 3. The sample mean, sample standard deviation and coefficient of variation of its length were 136.215 mm, 0.033 mm and 0.024%, respectively. The sample mean, sample standard deviation and coefficient of variation of its width were 64.636 mm, 0.018 mm and 0.027%, respectively. The sample mean, sample standard deviation and coefficient of variation of its height were 30.127 mm, 0.015 mm and 0.050%, respectively.

**Table 3.** Measured size of the gearbox base.

| Number of Times | Length (mm) | Width (mm) | Height (mm) |
| --- | --- | --- | --- |
| 1 | 136.232 | 64.641 | 30.157 |
| 2 | 136.183 | 64.644 | 30.125 |
| 3 | 136.238 | 64.598 | 30.138 |
| 4 | 136.167 | 64.653 | 30.117 |
| 5 | 136.272 | 64.648 | 30.143 |
| 6 | 136.248 | 64.625 | 30.118 |
| 7 | 136.181 | 64.619 | 30.126 |
| 8 | 136.215 | 64.658 | 30.130 |
| 9 | 136.209 | 64.643 | 30.102 |
| 10 | 136.204 | 64.638 | 30.129 |

The real length, width and height of the gearbox base are 136 mm, 64.5 mm and 30 mm. The dimensional deviations of length, width and height between the sample mean and the real value are 0.215 mm, 0.136 mm and 0.127 mm.

### 4.3. Summary

According to the algorithm in this paper, the slicing simulations and printing experiments of the lower teeth and gearbox base were carried out, respectively, and the surface roughness and size of the printed parts were measured, respectively. The surface roughness of the lower teeth printed by the algorithm in this paper was lower than that printed by the traditional slicing algorithm. Meanwhile, the dimensional accuracy of the gearbox base printed by the algorithm in this paper was higher than that printed by the traditional layered algorithm. The experimental results showed that this algorithm had higher dimensional accuracy and better surface quality than the traditional algorithm of some commercial software based on an STL model. This algorithm can be used to solve the problem of slicing contour distortion for some parts with the complex surfaces.

The dimensional accuracy and surface roughness of the same part printed under different printing conditions and using different printing equipment are obviously different. Therefore, the printing experiments in this paper were carried out on the cantilever AM platform, and the measurement results do not have wide universality. Nevertheless, the dimensional accuracy and printing quality of the parts printed using this algorithm were significantly improved compared with the traditional slicing algorithm. The algorithm in this paper can not only be applied to FDM additive manufacturing; it is also applicable to other kinds of additive manufacturing.

## 5. Conclusions

For the STEP model, a direct slicing algorithm was proposed, which avoided the process of STL file error diagnosis and repair, the conversion from a CAD model to STL file format, and the surface approximation of the STL triangular patch model to the original three-dimensional CAD model. This algorithm simplifies the process of AM data processing. For the parts with the basic types of surfaces—such as boundary curves, spherical surfaces and cylindrical surfaces—the traditional geometric method was used to calculate the intersection, which made full use of the special geometric properties of basic-type surfaces, improving the efficiency and robustness of the algorithm. For the parts with complex surfaces, a direct slicing algorithm based on a NURBS surface and STEP model was proposed. The NURBS surface was layered using a discrete tracking algorithm in order to determine the tracking starting point, obtain the intersection lines between the tangent plane and each NURBS sub-surface, and construct a closed layer contour.

Finally, the 3D model design, slicing simulations and printing experiments of the sample parts were carried out in order to verify the feasibility and accuracy of the slicing algorithm in this paper. By measuring the size and roughness of these printed parts, it was shown that the parts printed by the slicing algorithm in this paper had higher accuracy and better surface quality than the traditional slicing algorithm. This algorithm can not only be applied to FDM additive manufacturing; it is also applicable to other kinds of additive manufacturing.

**Author Contributions:** Conceptualization, X.H., Z.Z. and X.S.; methodology, X.H., Z.Z. and X.S.; software, X.H. and Z.Z.; validation, X.H. and Z.Z.; formal analysis, X.H. and X.S.; writing—original draft preparation, X.H. and L.C.; writing—review and editing, X.H. and L.C. All authors have read and agreed to the published version of the manuscript.

**Funding:** This research was supported by National Nature Science Foundations of China under Grant 51965014, Guangxi Nature Science Foundations of China under Grant 2018GXNSFAA050071, and National Nature Science Foundations of China under Grant 51705094.

**Institutional Review Board Statement:** Not applicable.

**Informed Consent Statement:** Not applicable.

**Data Availability Statement:** Not applicable.

**Acknowledgments:** The authors gratefully acknowledge the financial support from National Nature Science Foundations of China and Guangxi Nature Science Foundations of China.

**Conflicts of Interest:** The authors declare no conflict of interest.

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
