# Peer review of "An Additive Manufacturing Direct Slicing Algorithm Based on a STEP Model"

_electronics, doi:10.3390/electronics11101582_

Round 1
Reviewer 1 Report
Introduction
Can be improved by considering other methods of overcoming the limitations of slicers such as direct GCode scripting and then say what are the advantages and disadvantages of each method.
Page 2. Line 54: Please make sure you explain all the acronyms in the main text as well. Say what STEP stands for. Remember Abstract is separate from the main text. Check for other acronyms throughout the text.
Result and discussion
Page 7, line 306: I think this part needs more details in terms of what printing parameters you used including nozzle diameter, nozzle temperature and so on.
Page 7, line 312: How was the surface roughness measured? The values should have the standard deviation to understand the error of the measurement.
Page 8, line 325: Once again, the values for length, width and height do not have any standard deviation. How many did you measure?
Figures 5 and 6 should have scale bar.
Reviewer 2 Report
Dear Author,
Your manuscript entitled “An Additive Manufacturing Direct Slicing Algorithm Based on STEP Model” has gone under intensive review. During the review of your work, a few points have been observed that are listed below.
1) Some typographical errors have been observed.
2) Provide some basic information about fused deposition modeling.
3) Provide a separate section focusing on the significance of your work from the application point of view.
4) Some review related to the cost of the present proposed work w.r.to the existing system in terms of their industrial utilization is necessary.
5) Write a small summary before the conclusion.
6) Observed that in the whole manuscript, very old references have been cited. Use at least 60% of the recent references (between 2020 – till date).
The overall work is appreciable and the overall construction of the paper is very good.
Round 2
Reviewer 1 Report
The authors have addressed all comments. I am pleased to recommend it to publication.